# Vaccination Status and Attitudes towards Vaccines in a Cohort of Patients with Celiac Disease

**DOI:** 10.3390/vaccines10081199

**Published:** 2022-07-27

**Authors:** Andrea Costantino, Marco Michelon, Leda Roncoroni, Luisa Doneda, Vincenza Lombardo, Claudio Costantino, Maurizio Vecchi, Luca Elli

**Affiliations:** 1Gastroenterology and Endoscopy Unit, Foundation IRCCS Ca’ Granda Ospedale Maggiore Policlinico, 20122 Milan, Italy; vincenza.lombardo@policlinico.mi.it (V.L.); maurizio.vecchi@policlinico.mi.it (M.V.); luca.elli@policlinico.mi.it (L.E.); 2Department of Pathophysiology and Transplantation, University of Milan, 20122 Milan, Italy; marco.michelon@unimi.it; 3Department of Biomedical, Surgical and Dental Sciences, University of Milan, 20122 Milan, Italy; leda.roncoroni@unimi.it (L.R.); luisa.doneda@unimi.it (L.D.); 4Department of Health Promotion Sciences, Maternal and Infant Care, Internal Medicine and Excellence Specialties “G. D’Alessandro”, University of Palermo, 90145 Palermo, Italy; claudio.costantino01@unipa.it; 5Center for Prevention and Diagnosis of Celiac Disease, Foundation IRCCS Ca’ Granda Ospedale Maggiore Policlinico, 20122 Milan, Italy

**Keywords:** vaccines, vaccine hesitancy, celiac disease, coeliac disease, hyposplenism

## Abstract

(1) Background: The identification of vaccination status and attitudes towards vaccines among celiac disease (CD) patients is of great importance, but it has not yet been investigated. The aim of this study was to investigate coverage against vaccine-preventable diseases (VPDs), attitudes towards vaccinations, and its determinants among CD patients. (2) Methods: An anonymous web-based validated questionnaire was sent to a mailing list of CD adult patients. Patients were asked to self-report their previous vaccinations and attitudes towards vaccinations, which were defined as positive, negative, and partially positive/negative. The influencing factors towards vaccinations were investigated, and crude and adjusted odds ratios (AdjORs) with 95% confidence intervals (CIs) were calculated. (3) Results: The questionnaire was sent to 412 patients, with a response rate of 31.6% (130 patients, 105 women, median age 40 years, interquartile range 36–51). Patients self-reported vaccination against the following diseases: 73.8% tetanus, 42.3% flu, 20% measles, mumps and rubella, 19.2% meningitis, and 16.2% pneumococcus. Thirty-two people (24.6%) did not remember all of their previous vaccinations. In total, 104 (80%) respondents had a positive attitude towards vaccines, 25 (19.2%) a partially positive/negative one, and 1 a negative one. The determinants significantly influencing the positive attitude were being a graduate (AdjORs 7.49) and a belief in the possible return of VPDs with declining vaccination coverage rates (AdjORs 7.42), while the use of complementary and alternative medicines (AdjORs 0.11) and past negative experience (AdjORs 0.16) were associated with a negative attitude. (4) Conclusions: Despite four out of five CD patients showing a strong positive attitude towards vaccinations, one out of five had a partially negative one. Only a minority (16–20%) reported being vaccinated against some VPDs potentially harmful to their CD because of hyposplenism, such as meningitis and pneumococcus. The low vaccination rate against some VPDs, in spite of the 80% of CD patients stating a positive attitude towards vaccination, may be explained in part by patients’ vaccine hesitancy and in part by a possible role of physicians in under-prescribing vaccinations to these patients. These results may be a starting point for developing specific vaccination campaigns to increase vaccination rates against VPDs in CD patients.

## 1. Introduction

Infectious diseases and the potential for vaccination to prevent them represent a trending topic not only in the scientific community but also in public debate and sentiment [1]. Social media, with the power of its influence, have contributed in the last decade to bringing into vogue not only vaccine promotion but also medical misinformation with increased vaccine hesitancy [2]. According to the World Health Organization (WHO) Strategic Advisory Group of Experts on Immunization (SAGE), hesitancy is defined as “a delay in acceptance or refusal of vaccines despite availability of vaccination services” [3]; it represents a fundamental issue which has been deeply studied in order to improve the patient–doctor communication in promoting primary prevention and driving vaccine acceptance starting with family physicians [4].

The Coronavirus Disease 19 (COVID-19) pandemic, with the new global vaccination campaign and consequent spread of misinformation by anti-vaccination movements, has contributed to a resurgence of this issue [5].

Patients typically affected with chronic diseases such as diabetes mellitus, chronic obstructive pulmonary disease, and chronic kidney disease are elderly people with an increased vulnerability to infection; for this reason, they are generally encouraged to get vaccinated against vaccine-preventable diseases (VPDs) [6,7]. A recent post-pandemic survey among elderly Italian patients showed an optimal awareness of the protective measures given by vaccines [8]. However, celiac disease (CD), which is one of the most common chronic autoimmune disorders with a specific serological and histological profile triggered by gluten ingestion in genetically predisposed individuals, often affects young patients. It involves the small bowel and can cause malabsorption [9]; most patients respond to a gluten-free diet, while only a minority develops refractory disease requiring immunosuppressive therapies. [10] Even though it is unclear if CD patients have a generally increased risk of infectious diseases [11], they should generally be encouraged to receive all common vaccines against VPDs as the general population. Moreover, some pathogens could be harmful to CD patients. According to the European Society for the Study of Celiac Disease (ESsCD), CD can be associated with hyposplenism or functional asplenia, which could result in impaired immunity to encapsulated bacteria, with an increased risk of such infections. For this reason, CD patients who are known to be hyposplenic should be administered at least the pneumococcal vaccine [12]. However, the ESsCD states that it is not clear whether vaccination with the conjugated vaccine is preferable in this setting and whether additional vaccination against Haemophilus, meningococcus, and influenza should be considered if not previously given [12]. However, Mårild et al. suggested considering additional vaccination against influenza because of an observed increased risk of hospital admission for this infection in CD patients [13].

To the best of our knowledge, there is no evidence that patients with CD are more fearful about vaccination compared with the general population and if they are aware of vaccinations already received and ones that could be offered to them. For this reason, the identification of vaccination status and the attitudes toward vaccines of CD patients is of great importance. 

Knowledge of the factors influencing the attitudes could help drive effective doctor–patient communication and patient association campaigns to improve vaccination among CD patients

## 2. Materials and Methods

### 2.1. Study Design

In February 2021, an anonymous web-based adapted version of a validated questionnaire [14] was sent twice in 10 days to a mailing list (the only with expressed permission to receive email campaigns) of 412 celiac patients from the “Ioeilglutine Onlus” (the nonprofit organization devoted to biomedical research of gluten disorders related to Our Center) and of patients followed the Center for Prevention and Diagnosis of Celiac Disease of the Foundation IRCCS Ca’ Granda Ospedale Maggiore Policlinico of Milan, Italy, where approximately 3000 patients are followed-up. 

Patients were asked to self-report their previous vaccinations and give their attitudes towards vaccinations (with a multiple-choice test), which were defined as positive, negative, and partially positive/negative. The influencing factors towards vaccinations were investigated, and crude and adjusted odds ratios (AdjORs) with 95% confidence intervals (CIs) were calculated. 

The questionnaire was adapted to patients with CD from a previously validated questionnaire on vaccine hesitancy [15] and sent to patients as a URL link in an email. It was developed online using the EUSurvey platform by our center. Completion of the web-based survey did not result in any benefit or financial compensation for respondents. The questionnaire investigated three areas: sociodemographic data, CD-related and lifestyle data, and attitude to vaccinations. 

The questionnaire was divided into seven sections (Table 1).

### 2.2. Statistical Analysis

Data were automatically collected on EU-Survey. The sample size was calculated assuming the percentage estimated of positive and not positive attitude towards vaccines; accordingly, 110 respondents would be needed. Absolute and relative frequencies were calculated for the categorical (qualitative) variables, and quantitative variables were summarized by their means and range. All variables found to have a statistically significant association with vaccination attitude in the univariate analysis were included in a multivariate backward stepwise logistic regression model. All variables with a *p* value ≤ 0.20 were selected in the multivariate model to guarantee a more conservative approach. A backward stepwise regression model was used. The crude odds ratio (crude OR) and the adjusted OR (AdjOR) with 95% confidence intervals (CIs) were also calculated in the logistic regression model. The level of significance chosen for the multivariate logistic regression analysis was 0.05 (2-tailed). We entered all the information into a database created with Epi Info™ 3.5.4 (Centers for Disease Control and Prevention, Atlanta, GA, USA). All the data were analyzed using the statistical software package Stata/MP 12.1 (StataCorp LP, College Station, TX, USA).

### 2.3. Ethics

The study was approved by the local Ethics Committee of Foundation IRCCS Ca’ Granda Ospedale Maggiore Policlinico, Milan (approval no. 1527/2020). All the subjects received an email explaining the rationale of the study and the digital informed consent to participate. As they agreed, the subjects were directed via a link to an online structured questionnaire on the EU-Survey platform (https://ec.europa.eu/eusurvey/, accessed on 24 May 2022) supported by the European Commission, which allows collecting sensible data with no user identification via IT tracking, profiling cookies, or geographical location or personal/sociodemographic/health data.

## 3. Results

### Questionnaire Analysis

One hundred and thirty patients answered the questionnaire, with a response rate of 31.6%. The baseline characteristics of the study population, inclusive of sociodemographic, occupational, and lifestyle status, are listed in Table 2. A total of 130 patients (80.8% women) answered the web-based questionnaire, while the mean age was 40 years (interquartile range 36–51). Most of the patients in the study were graduates (55.4%); all patients were affected by CD (100%); the majority were diagnosed more than 10 years before (58.5%); a minority (7.7%) worked as healthcare professionals.

Table 3 lists the previous vaccinations received by the study population. A total of 73.8% of the sample respondents were vaccinated for tetanus, while 19.2% and 16.2% were vaccinated for meningitis and pneumococcus, respectively. Thirty-two respondents (24.6%) could not remember all their previous vaccinations received. 

Attitudes towards vaccinations are listed in Table 4. Eighty percent of the respondents stated they had a positive or a partially positive/negative (19.2%) attitude towards vaccinations, while one patient expressed a negative position. One hundred and twenty patients (92.3%) reported a willingness for future vaccinations; 112 out of 130 patients answered the question about the willingness to vaccinate their children in the future. Of these, 83% expressed the intention of totally vaccinating their children, while 15.2% stated they would partially vaccinate their children; only two showed reluctance about childhood vaccinations.

One hundred and sixteen (89.2%) believed that the decline of vaccination coverage would result in a possible return of VPDs; 129 patients out of 130 believed that vaccinations alone (38.5%) or in association with other preventive strategies (60.8%) were the best way to prevent VPDs; one patient reported that others (e.g., diet, homeopathy, physical activity) were the best strategies to prevent VPDs. One hundred and fifteen (88.5%) of the patients reported that CD was not the motivating factor leading to previous vaccinations. Twenty-five patients (19.2%) self-reported a previous negative experience with vaccines (whether personal or not). Finally, 94.6% of the patients displayed higher confidence in the information about vaccines provided by healthcare professionals than compared to mass media.

Table 5 reports the results of the univariate/multivariate analysis. The determinants positively influencing attitudes towards vaccinations were being a graduate (AdjOR 7.49, 95% CI: 1.74–32.1, *p* < 0.01) and a belief in the possible return of VPDs with declining vaccination coverage rates (AdjOR 7.42, 95% CI: 1.32–41.6, *p* < 0.05), while the use of complementary and alternative medicines (CAM) (AdjOR 0.11, 95% CI: 0.01–0.71—reciprocal value AdjOR 9.09, 95% CI: 1.41–100.0, *p* < 0.05) and past negative experience (AdjOR 0.16, 95% CI: 0.04–0.62—reciprocal value AdjOR 6.25, 95% CI: 1.61–25.0, *p* < 0.01) were negatively associated with vaccinations.

## 4. Discussion

### 4.1. Vaccine Hesitancy in General Population

Vaccine hesitancy is a global, complex, and ever-changing phenomenon, and it represents one of the most important criticisms in public health today. Even though it would appear to be a contemporary discussion, the public debate on vaccination is a deeply rooted phenomenon in Western culture. The first law about mandatory vaccination in Europe was enacted in 1853 in the United Kingdom (Vaccination Act 1853), which gave rise to violent opposition and the Victorian anti-vaccination movement [16]. Italy has always given primary prevention a central role in its healthcare system and has a long legislative history of mandatory childhood vaccination. In 1939, compulsory vaccination for diphtheria was applied with Italian Law n. 891/1939; anti-tetanus and anti-poliomyelitis vaccinations were made mandatory in 1967 (Italian Law n. 1518/1967), while hepatitis B vaccination became compulsory in 1991 (Italian Law n.165/1991) [17,18]. Since 2010, the debate around vaccines in Italy has gained new attention after the drastic decline in the measles vaccine rate, which resulted in a severe outbreak of measles in 2017 with 4991 cases and 4 deaths [19,20]. The Italian government reacted to the re-emergence of a VPD by reintroducing mandatory vaccinations for children aged 0–16 in order to attend school, which caused political and social opposition led by vaccine-hesitant groups [21].

Vaccine hesitancy may be attributed to three prominent factors: safety concerns, negative stories, and personal knowledge. The safety of a potential COVID-19 vaccine is a major concern, even for those who are very willing to have it. Some patients may feel reassured by patient–doctor effective communication [5]. A major issue for people is how quickly a vaccine would have been produced and that doctors and pharmaceutical companies would not know all the side effects. Some hesitant may feel confused by the negative stories about it rather than being resolutely against it [5]. Vaccine hesitancy is often determined by incorrect beliefs about health, diseases, and vaccines, which may have been influenced by misinformation. The mass media’s emphasis on the hypothetical side effects of vaccines has triggered waves of misinformation on the safety of vaccines, mainly concerning long-term side effects, the toxicity of adjuvants and preservatives, and the weakening of the immune system [1,2].

A first attempt at framing this multifaceted situation was proposed in 2000 by a study taken up by the European Center for Disease Control and Prevention (ECDC), which distinguishes patients into different categories [21]:Hesitant: concerned about the safety of vaccines and unsure about needs, procedures, and timetables.Disinterested: with little awareness of vaccination (considered a low priority) and inadequate perception of the risk of preventable diseases.Excluded: disadvantaged with limited or difficult access to treatment for social, economic, and integration reasons.Anti-vaccinationists: with an attitude of rejection and active resistance due to personal, cultural, and religious convictions.

### 4.2. Impressions on Vaccinations Status and Attitudes among Patients with Celiac Disease

To our knowledge, this survey is one of the first studies reporting vaccinations and attitudes towards vaccinations among patients with CD. Data reported in Table 3 show that only a limited number (three-quarters) of the CD patients could remember their vaccinal status. The most reported vaccination received is against tetanus; considering that this vaccination has been mandatory in Italy since 1967 [17], the fact that only 73.8% of patients declared themselves to be so vaccinated is likely to be an underestimation of the real vaccinal status of this cohort. Supporting this, 24.6% of the patients could not remember previous vaccinations, which highlights a poor general focus on the importance of primary prevention and emphasizes the need for developing significant patient–doctor communication. Even more relevant is the fact that less than 20% of the patients reported being vaccinated against encapsulated bacteria.

Currently, there are no data on the rate of CD patients vaccinated against pneumococcus, though it would appear that pneumococcal vaccination in adult CD patients is considerably underused [22]. Similar to our results, Khan et al., in a 2013 study that analyzed 119 CD patients <65 years with at least one comorbidity, found that only 19.2% of the sample patients had been vaccinated against pneumococcus [23], showing that the real vaccination status in the CD population has not changed much in the last 10 years.

As previously demonstrated, hyposplenism prevalence in CD patients varies from 19% in the uncomplicated CD without the autoimmune disease to 80% in cases associated with premalignant or malignant lesions. Moreover, splenic hypofunction is not related to gluten-free diet duration, and its prevalence increases when CD patients are also diagnosed with autoimmune disorders, such as autoimmune thyroiditis or insulin-dependent diabetes [24,25]. This association gains relevance insofar as hyposplenic patients have an increased risk of developing severe infections with encapsulated bacteria such as Streptococcus pneumoniae, Neisseria meningitides, and Haemophilus influenzae type b, as well as gram-negative bacteria such as Capnocytophaga canimorsus [26]. Furthermore, William et al. demonstrated that CD is even the most frequently associated disease with functional hyposplenism [27].

Due to the potentially harmful diseases associated with encapsulated bacteria in CD patients, many studies have been conducted, in particular on the pneumococcal vaccine. Simons et al. stated that the pneumococcal vaccine should be considered for CD patients, with particular attention to those aged 15–64 years and never vaccinated before, as CD is associated with an increased risk of *S. pneumoniae* infection [28]. Other studies suggest vaccinating CD patients against pneumococcus in those with advanced age at diagnosis, concomitant autoimmune disorders, complicated CD, previous history of major infection/sepsis, and/or thromboembolism and splenic atrophy [29]. The potential major risk of infection in CD patients related to hyposplenism [29,30] reinforces the importance of optimizing efficient patient–doctor communication.

Data in Table 4 show a generally positive attitude towards vaccinations, despite one out of five patients expressing a partially negative view, and the willingness to get vaccinated in the future (even against COVID-19) also revealed a positive attitude towards vaccinations. It is of great interest that the majority of the respondents (88.5%) said that CD did not motivate them to receive their previous vaccinations, which shows a lack of the perceived risk of contracting infectious disease due to their CD. This is also of great relevance insofar as CD is a chronic autoimmune illness that can require immunosuppressive therapy for complicated cases, even if in a small minority [8], with its consequent increased risk of infectious diseases. Moreover, the belief held by 90% of the patients about the possible return of VPDs with the decline of vaccination coverage is of great importance. It demonstrated a significant positive association with vaccination attitude, showing, in contrast to the low rate of vaccinations that emerged in our study, the awareness of the importance of vaccination campaigns.

As emerged in other studies, a positive attitude to CAM had a significant negative association with attitude towards vaccinations [31], which has been previously reported. CAM users may believe that vaccines and other drugs commonly prescribed by physicians are harmful and instead use alternative medicines and practices such as chiropractic and acupuncture (also when there is a lack of evidence). It is presumed that CAM practitioners advise their clients against vaccines [32].

Most of the respondents trust healthcare professionals more than mass media for providing information on their healthcare: this is relevant for awareness campaigns and the increase in vaccinations among CD patients that could be achieved after persuasive communication. Of great importance is also the role of general practitioners, as they are often the first referring physician for CD patients: in fact, they should ensure that CD patients are fully vaccinated, if necessary, against encapsulated bacteria [33]. The physician (whether the specialist or the general practitioner) can also play a pivotal role regarding vaccination in contrast to mass media or patient association campaigns. Moreover, more time and effort should be spent on trying to convince non-graduate patients. Healthcare professionals should also investigate the reported negative previous experiences with vaccinations in order to value their real burden or if they were common side-effects or just nocebo effects; we demonstrated that these negative experiences significantly influenced their attitudes towards vaccinations.

### 4.3. Strengths and Limitations

There are some limitations to the present study. First, a possible response bias could be considered as patients who responded to the questionnaire could be more predisposed to vaccines than those who did not. Since the questionnaire was anonymous, we do not have the possibility to identify and distinguish between those who responded and those who did not. 

A possible limitation is that the questionnaire was sent to a mailing list of only 412 with expressed permission to receive email campaigns out of about 3000 patients followed-up in our Center, with a possible selection bias of those more interested in news or research about their disease or with a more proactive behavior. Another limitation is that 80.8% of people answering the survey were women. It is well-known that women are generally more interested in health and actively seek health-related information rather than men [34], even though there was no significant association in our study between sex and attitude towards vaccinations. Finally, our results were obtained from a self-reported questionnaire; moreover, 24.6% could not remember their previous vaccination status. These two are the major limits of self-reported questionnaires on vaccinations compared to vaccination cards or any official records. Nevertheless, a questionnaire is the most effective way to investigate patient vaccination status in a short period while investigating so many parameters. 

Previous studies aimed to examine the accuracy of self-reported vaccination status compared with official records [35,36]. A study showed a limited validity of self-reported reports for HPV vaccine uptake (with a positive predictive value of 87.7% but a negative predictive value of 54.5%;) [35]. This low NPV may lead to a reduced estimation of vaccination coverage. Another study showed that among all participants with electronic documentation of smallpox vaccination, 90% self-reported being vaccinated; of all participants with no electronic record of vaccination, 82% self-reported not receiving a vaccination. Therefore, the overall k statistic indicated a substantial and acceptable agreement (k = 0.62) [36]. 

A final limit is given by the numerosity of respondents (130), but the questionnaire had many questions; it was voluntary and without any compensation. Future multicenter studies, possibly sent by one of the patients’ associations, may show a broader national picture of the vaccination status of patients with CD. 

On the other hand, our study has many strengths. It is the very first study to analyze the attitudes towards general vaccinations in patients affected by CD, which represents one of the most common chronic illnesses. Second, our study included patients not followed by our tertiary referral center, as part of the respondents, some Ioeilglutine Onlus members, are not followed up regularly. Therefore, it represents a realistic picture of the vaccination status in the CD population; hence, there is no selection bias in those patients who regularly followed up and correctly advised on the importance and possibility of getting vaccinated. Moreover, the questionnaire was validated. We first investigated all the factors influencing their attitudes towards vaccinations, and the response rate was comparable to the same web-based questionnaire (~25%) [37,38].

Future research should investigate vaccination status through vaccination reports and clinical medical records in a big multicenter, regional or national cohort of patients with CD. Big data relating to disease exemption codes and vaccination data could be easily analyzed. However, there may be possible issues related to patients’ privacy concerns. 

### 4.4. Possible Strategies to Improve Vaccination Coverage among Celiac Patients

Previous studies and this study may suggest some strategies that could be adopted to improve vaccine uptake among patients with CD through the implementation of certain approaches:Improving the involvement for vaccine prescription of HCPs that regularly care for CD patients, such as gastroenterologists, internists, general practitioners, and nutritionists.Promoting guidelines that provide specific indications on how to actively call for vaccination, especially in relation to young, adult, and elderly patients with CD/CD refractory disease and other chronic gastrointestinal diseases.Spreading of vaccination culture, through the diffusion of the vaccination message to all patients, with the help of patients’ associations.Better involvement of CD patients in vaccination practice through the implementation of awareness campaigns aimed at adolescents, young adults, and adults and through the administration of vaccines within vaccination campaigns or the organization of specific vaccination events [39,40,41,42,43,44].Optimizing the patient–doctor communication for those patients with a higher probability of being hesitant against vaccines (e.g., CAM users, patients with a previous personal experience, lower educational level).

Education represents a better way to improve vaccinations among CD patients. In fact, the discrepancy between the real danger and the perceived risk of VPDs can lead to inappropriate behavior that does not comply with the public health measures recommended for both the general population and at-risk cohorts, such as patients with chronic gastrointestinal diseases. Even when a sufficient level of knowledge is present, messages issued by the HCPs and effective warnings seem to be necessary. Digital instruments could be a valuable tool for promoting vaccination campaigns. 

Considering the education of professionals, original articles, systematic reviews, or meta-analyses on vaccines administered in all cohorts of the population (with particular attention to patients with chronic disease (among which CD is one of the most prevalent chronic immune-mediated diseases) as well as strategies adopted to promote vaccination adherence among these categories are necessary today for the scientific community [45].

European guidelines suggest specific recommendations that CD patients should follow a regular vaccination schedule, regardless of age at diagnosis [12]. Other guidelines do not give indications about vaccinations among patients with CD [46,47]. Therefore, it has already been proposed that in the next future, specific clinical practice guidelines for vaccination programs among patients with CD will be necessary [22].

## 5. Conclusions

Even though four out of five CD patients showed a strong positive attitude towards vaccinations, one out of five had a partially negative view. Moreover, only a minority (~20%) reported being vaccinated against some VPDs potentially harmful to their chronic illness. This may suggest a possible role of physicians in prescribing not properly vaccinations to this population since 80% of patients have a positive attitude. 

These results may be a starting point for developing specific vaccination campaigns to increase vaccination rates against VPDs in CD patients. HCPs should investigate better vaccination status among CD patients, encourage them to receive all common vaccines and some of them to get vaccinated against other possibly serious VPDs. 

With regard to the hesitant patients, the identification of the determinants influencing patient attitude towards vaccinations may help to optimize patient–doctor communication and develop specific vaccination campaigns. 

## Figures and Tables

**Table 1 vaccines-10-01199-t001:** Sections and main items of the vaccination questionnaire.

A	Sociodemographic information including gender, age, nationality, level of education, marital status, parental status, and work activities (specifically if healthcare providers).
B	Information regarding the course of celiac disease, in terms of disease classification (e.g., refractory celiac disease, non-celiac gluten sensibility), disease duration, therapies, and adherence to the gluten-free diet.
C	Lifestyle, health-related behaviors, and attitudes, including smoking, physical activity, and approach to screening services.
D	Knowledge and perceptions regarding vaccination and vaccine-preventable diseases.
E	Vaccination history.
F	Sources of information on vaccines such as general practitioners, the mass media, and pharmacists.
G	Reports by people close to the respondent regarding vaccines and adverse events.

**Table 2 vaccines-10-01199-t002:** Sociodemographic, occupational, and lifestyle characteristics of the study population (n = 130).

Characteristics	n (%)	CI 95% (%)
**Gender**		
Male	25 (19.2)	13–27.3
Female	105 (80.8)	72.7–87.0
**Age (years), mean (range)**	40 (36–51)	
**Marital status**		
Married/cohabitant	92 (70.8)	62.7–78.9
Single/divorced/widowed	38 (29.2)	21.1–37.3
**Educational level**		
Undergraduate	58	44.6
Graduate	72	55.4
**Number of family members**		
≤2 members	49	37.7
>2 members	81	62.3
**Disease**		
Celiac disease	130 (100)	N.V.
**Working as healthcare professionals**		
No	120	92.3
Yes	10	7.7
**Profession**		
Manager/entrepreneur/freelancer	28	21.5
Employee/technical profession	82	63.1
Manual work/craftsman	7	5.4
Student/housewife/elderly/unemployed	13	10
**Alcohol intake**		
No	57	43.8
Yes often/minimal consumption	73	56.2
**Self-reported active lifestyle**		
No	48	36.9
Yes	82	63.1
**Vegetarian or vegan diet**		
No	123	94.6
Yes	7	5.4
**Smoking habit**		
Non-smoker	114	87.7
Smoker/ex-smoker	16	12.3
**Use of complementary and ** **alternative medicines**		
No	118	90.8
Yes	12	9.2
**Years from diagnosis**		
<5 years	16	12.3
5–10 years	38	29.2
>10 years	76	58.5

**Table 3 vaccines-10-01199-t003:** Previous vaccinations among the study population with CD (n = 130).

Previous Vaccines	n (%)	CI 95% (%)
Tetanus	96 (73.8)	67.5–78.9
Flu	55 (42.3)	38.5–45.5
Measles, mumps, and rubella (MMR)	26 (20.0)	16.4–25.6
Meningitis	25 (19.2)	14.7–23.2
Pneumococcus (PCV13 or PPSV23)	21 (16.2)	13.5–19.8
Patients who did not remember previous vaccines	32 (24.6)	19.2–28.7

**Table 4 vaccines-10-01199-t004:** Attitudes towards vaccinations in the study population (n = 130).

	n (%)	CI 95% (%)
**Attitudes towards vaccinations**		
Negative	1 (0.8)	0.3–2.5
Positive	104 (80.0)	74.2–86.5
Partially positive/negative	25 (19.2)	15.8–22.6
**Willingness to get vaccinated again in the future**		
No	10 (7.7)	5.1–9.4
Yes	120 (92.3)	88.5–95.2
**Willingness to vaccinate your children in future (n = 112)**		
No	2 (1.8)	0.6–3.4
Yes, totally	93 (83.0)	78.6–86.7
Yes, partially	17 (15.2)	12.4–18.9
**Belief in possible return of VPDs with** **Decline of vaccination coverage**		
No	14 (10.8)	7.9–13.5
Yes	116 (89.2)	84.5–93.6
**Best strategy to prevent VPDs**		
Vaccination	50 (38.4)	34.6–41.9
Vaccination + other strategies	79 (60.8)	57.2–63.4
Other (diet, physical activity, homeopathy, etc.)	1 (0.8)	0.4–2.2
**Celiac disease/ongoing therapy as motivation for previous vaccinations done**		
No	115 (88.5)	82.6–91.8
Yes	15 (11.5)	8.6–14.6
**Previous negative experiences with vaccines** **(personal/family members/relatives reported/referred)**		
No	105 (80.8)	76.5–84.6
Yes	25 (19.2)	16.3–23.4
**Higher confidence in vaccines from healthcare** **professionals compared to mass media**		
No	7 (5.4)	3.2–7.8
Yes	123 (94.6)	91.3–97.6

**Table 5 vaccines-10-01199-t005:** Crude odds ratio (OR) and adjusted OR (AdjOR) for sociodemographic, lifestyle, and clinical characteristics, knowledge, attitudes, and perceptions about general vaccination, with positive attitudes towards vaccines in patients with celiac disease.

	Crude OR	CI 95%	*p* Value	AdjOR	CI 95%	*p* Value
**Gender**						
Male	ref		0.56			
Female	0.71	(0.22–2.28)				
**Age in years (continuous variable)**	0.99	(0.96–1.03)	0.73			
**Marital status**						
Single/divorced/widowed	ref		0.25			
Married or cohabitant	0.53	(0.18–1.52)				
**Children <10 years of age**						
No	ref		0.71			
Yes	1.29	(0.34–4.82)				
**Educational level**						
Undergraduates/non-graduates	ref		<0.05	ref		<0.01
Graduate	2.21	(1.12–5.39)		7.49	(1.74–32.1)	
**Working as healthcare professionals**						
No	ref		0.22			
Yes	0.37	(0.08–1.68)				
**Adherence to gluten-free diet**						
No	ref		0.35			
Yes	3.90	(0.39–35.2)				
**Smoking habit**						
No	ref		0.89			
Yes	0.65	(0.24–1.45)				
**Self-reported regular physical activity**						
No	ref		0.52			
Yes	1.33	(0.55–3.19)				
**Alcohol intake** **(2–3 units/day or more)**						
No	ref		0.79			
Yes	0.89	(0.47–1.89)				
**Use of complementary and alternative medicines**						
No	ref		<0.05	ref		<0.05
Yes	0.13	(0.04–0.48)		0.11	(0.01–0.71)	
**Adherence to other preventive measures** **(e.g., oncological screenings)**						
No	ref		0.36			
Yes	0.51	(0.14–1.88)				
**Willingness to be vaccinated in future** **(even against COVID-19 and other vaccines)**						
No	ref		<0.001	ref		<0.01
Yes	10.3	(3.91–27.1)		8.78	(2.40–32.1)	
**Considering possible return of VPDs with decline of vaccination coverages**						
No	ref		<0.01	ref		<0.05
Yes	6.22	(1.87–20.7)		7.42	(1.32–41.6)	
**Previous negative experience with vaccinations** **(personal, family members, relatives, or also reported/referred)**						
No	ref		<0.01	ref		<0.01
Yes	0.21	(0.08–0.55)		0.16	(0.04–0.62)	
**Higher confidence in vaccines in healthcare professionals compared to mass media**						
No	ref		0.14	ref		0.09
Yes	3.26	(0.68–15.6)		6.54	(0.76–56.1)	

## Data Availability

The datasets generated during and/or analyzed during the current study are not publicly available but are available from the corresponding author upon reasonable request.

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
