# Peer review of "Vaccination Status and Attitudes towards Vaccines in a Cohort of Patients with Celiac Disease"

_vaccines, 2022, doi:10.3390/vaccines10081199_

Round 1

Reviewer 1 Report

The authors are thanked for the effort to do research work of great interest to readers, especially in the context of the COVID pandemic, to understand the factors related to positive or negative attitudes to vaccination. However, it is possible to identify some biases such as participation, memory, and the selection of intervals for the Likert scaling methodology. The authors must deepen and discuss more the limitations, the biases and their impact on the results, the interpretations of the same, and their impact on the study's conclusions.

Analysis. In the multivariate regression model, how did you test interactions and effect modifications, and how did you decide to keep the variables in the final model?

Results. The response rate was 31%. Are there significant differences between participants and non-participants? A bivariate analysis compared the differences at least in age and sex. They can identify a selection bias and its implications for the results and conclusions to support this statement “; hence there is no selection bias of those patients 269 regularly followed up and correctly advised on the importance and possibility of getting 270 vaccinated.”

Table 2. The foot graph of the figure is redundant with table 3, and the information in Table 2 is out of context.

Attitudes towards vaccination used a Likert scale with three responses, and it is known that most respondents will choose the middle answer. Is there a possibility of bias, and how would authors interpret it?

I suggest changing the direction of the original question to invert the OR, or report the reciprocal value, so the readers should consider that aOR of 0.13 (95%CI (0.14-1.88) will be 7.69 (95%CI 25-2.08) if it is interpreted as a risk factor.

24.6% could not remember the vaccination status; there is a memory bias and would have affected the results and how to affect the “fact that less than 20% of the patients reported 193 to be vaccinated against encapsulated bacteria.”. Authors must discuss the differences between using a self-reported vaccination status and vaccination cards or any official records. As far as I know, some studies have addressed the difference between methods.

Use of complementary and alternative medicines and Previous negative experience with vaccinations (personal, family members, relatives, or also reported/referred) are two primary variables related to non-vaccinated persons. Authors must improve the discussion concerning these factors because of their relevance.

An additional suggestion:

Please include in table 2 and 3 the 95% confidence interval. Thanks to all authors.

Author Response

We thank the reviewer for their precise comments. 

As regards the first precise observation, a backward stepwise regression model was used (we added in the Analysis paragraph). 

As regads the second point, as the questionnaire was anonymous we cannot identify the respondents and the non respondents. We added this as a limit of the study. Thank You. 

As regardas Table 3 and the text we kept both text and table to have a more visual impact and to show 95%CIs that were added as you properly suggested, thanks. 

As regard the reponse to the Vaccination Attitude, it was not a Likert Scale but a multiple choice in which patients have to choose the definition that could fit more. We thank the Author for his/her comment and we added to the text. 

As regards the direction of the original question, we inverted the OR in the text as you properly suggested. 

As regards the possibile differences between self-reported vaccination status and official records, we added this in the discussion and added some previous study investigating this. Thank You for your suggestion. 

As the regards the potential role of CAM and previously negative experiences among relatives/friends, we added a paragraph in the disccussion. 

As regards the last point, we added 95% CI in the table 2, 3 an 4 

Thank You very much, once again, for Your precise corrections of our Manuscript. 

Best Regards

Reviewer 2 Report

The work done by  entitled "Vaccination Status and Attitudes towards Vaccines among Patients with Celiac Disease "need some modification before being accepted for publication.

At first the introduction need to be revised and rewritten clearly, for example, adding some sections for the different kidney diseases and also some sentences for the previous reports mentioned similar or different situations.

Table 1 is not clear; the sociodemographic data need to be more organized,, please explain as mentioned in your questionnaire.

Same are required for table 2

In discussion, Vaccine Hesitancy in General Population , this need to be discussed in a better way in correlation to other vaccine hesitancy studies

Author Response

We want to thank the reviewer for their very precise comment. 

As suggested for the discussion, we added more detailed and precise paragraphs about vaccine hesitancy among the general population, in correlation to other studies, in the discussion. 

We made the Table 1 clearer and added 95%CI in the first tables. 

Thank You once again for your precise comments and suggestions, 

Best regards. 
